# Investigating the Age-Related Association between Perceived Motor Competence and Actual Motor Competence in Adolescence

**DOI:** 10.3390/ijerph17176361

**Published:** 2020-09-01

**Authors:** Conor Philpott, Brian Donovan, Sarahjane Belton, Diarmuid Lester, Michael Duncan, Fiona Chambers, Wesley O’Brien

**Affiliations:** 1School of Education, Sports Studies and Physical Education Programme, University College Cork, T12 KX72 Cork, Ireland; 114321861@umail.ucc.ie (B.D.); 111724699@umail.ucc.ie (D.L.); f.chambers@ucc.ie (F.C.); wesley.obrien@ucc.ie (W.O.); 2School of Health and Human Performance, Dublin City University, D09 Y5N0 Dublin 9, Ireland; Sarahjane.belton@dcu.ie; 3Research Centre for Applied Biological and Exercise Sciences, Coventry University, Priory Street, Coventry CV1 5FB, UK; michael.duncan@coventry.ac.uk

**Keywords:** functional movement, fundamental movement skills, motor competence, cognition, motor skills

## Abstract

Irish adolescents have been found to possess high perceptions of motor competence. However, there is an evidential value to investigating the strength of the relationship between adolescent perceptions of motor competence and their low levels of actual motor competence. The purpose of this research was to gather data on the fundamental, functional, and perceived motor competence in adolescents, differentiated by year group, to discern if participants could assess their perceptions of ability. Data were collected on adolescents (*N* = 373; mean age: 14.38 ± 0.87 years; 47.7% female) across six second-level schools in Ireland, including measurements of fundamental movement skills, functional movement, and perceived motor competence. Poor levels of fundamental and functional movement were observed, with significant differences between year groups detected. Participants in 1st year scored the highest in overall fundamental movement skills; however, for overall functional movement, 3rd-year participants scored highest. High levels of perceived motor competence were reported across the entire sample. These scores did not align with actual motor competence, nor did any alignment between these measurements improve with aging, countering theorized age-related associations. Future research should target low levels of actual motor competence while emphasizing the cognitive aspects of movement to ensure greater accuracy between actual and perceived motor competence.

## 1. Introduction

Global adolescent physical activity (PA) levels remain at disconcertingly low levels, with few signs of future improvements [1]. As low PA levels are associated with a higher degree of obesity and an increased risk of non-communicable diseases, increasing PA levels must become a vital part of future initiatives to improve the health status of future generations [2]. Irish PA participation levels for children and youth remain perilously low, with only 13% meeting current recommendations of 60 min of PA per day [3]. Childhood and adolescent PA patterns typically subsist throughout the lifespan, clearly underlining the importance of establishing an active lifestyle as early as possible [4]. Both fundamental movement skills (FMS) and perceived motor competence (PMC) are recognized as positive pillars that can form a critical role in boosting adolescent PA levels [5].

Fundamental movement skills (FMS) are frequently cited as the ‘building blocks’ of intricate movements, which are necessary for participation in sports, games, and PA [6,7]. FMS are comprised of three key subsets: locomotor skills that involve the transportation of the body in any direction (e.g., vertical jump) [8], object control, which requires the manipulation or control of a projectile [8], and balance. Positive associations between FMS performance and cardiorespiratory fitness have been found repeatedly across systematic reviews [9,10]. An inverse relationship between FMS performance and weight status has also been observed with greater FMS ability associated with reduced weight [10,11]. Greater levels of FMS performance are associated with both increased PA time and sustained engagement in PA throughout childhood and adolescence [8,12], which suggests that FMS form a vital part of the rounded skill set necessary for students to prosper in PA throughout their lifespan [8,12,13].

Global FMS figures vary by region, but the commonality found across countries is that of poor proficiency. Among a New Zealand childhood cohort (*n* = 701) aged between 0 and 8 years old, less than half of children demonstrated proficiency in the skills of kicking, throwing, and striking [14]. In a Brazilian study (*n* = 2000), less than half of children between the ages of 3 and 10 years old demonstrated mastery across locomotor skills, and the object-control subset measured object-control skills (strike, stationary dribble, catch, throw, underhand roll) [15]. A European cross-cultural study indicates that 10% and 40% of children in northern and southern Europe respectively, possess poor levels of movement proficiency [16].

Both Irish childhood and adolescent research studies indicate subpar FMS levels [17,18]. One Irish study indicated that just over 50% of Irish children between the ages of 5 and 12 years old (*n* = 2048) displayed proficiency across locomotor and object-control skills [19]. A ceiling effect of FMS performance may occur among Irish cohorts, which may further explain the findings of Lester et al. (2017). Among Irish adolescents, less than 40% of participants displayed proficiency in two locomotor skills (vertical and horizontal jump) and four object-control skills (kick, dribble, strike, throw) by the end of their third year of second level school (ages 15–16 years old), indicating a stagnation in skill development in adolescence [20]. The low levels of physical education (PE) participation among students in Irish primary school settings, in addition to a lack of specialist PE teachers at the primary school level, may be a critical factor in the low skill levels reported in childhood and adolescence [21]. Insufficient pedagogical knowledge from non-specialist teachers and minimal PE time may negatively be impacting on FMS development throughout childhood and into adolescence in Irish students, with poorer quality PE negatively impacting on skill acquisition [21]. Furthermore, a lack of time in PE may undermine FMS development in the primary years, with 18% of Irish children receiving less than 30 min of PE per week [3]. To nurture both FMS proficiency and PA levels, both functional movement and perceived motor competence could be deemed essential ingredients for consideration [22].

Functional movement refers to the capacity to move with adequate muscle and joint functioning [23]. While research on functional movement and its association with PA or perceived motor competence (PMC) is limited, an association between FMS and functional movement is theoretically informed. Critical elements of functional movement assessments, including shoulder mobility and postural stability, are essential to a higher level of performance in a number of FMS, such as vertical jumping, running, and throwing [24,25,26]. Motor competence (MC) has been defined as a term to describe goal-directed human movement [27]. Both functional movement and FMS fit this description, and their associations indicate that assessing both elements in an assessment battery would be highly pertinent. By diversifying both the range of movements assessed and assessment batteries used in MC testing through the inclusion of functional movement, a more rounded perspective of MC would be gathered, which reflects the greater diversity of PA opportunities available to an individual across their lifespan [22,28].

The Functional Movement Screen^TM^ (FMS^TM^) is the pre-eminent assessment tool within the field of functional movement patterns, and this instrument seeks to detect asymmetries or pain when performing these critical movements [29]. Irish studies indicate that adolescents possess poor levels of functional movement [20,22]. These figures are lower than previously cited global figures found among Indian adolescents [30] and in American populations [23,31]. These Irish results are marginally above the well-established critical composite score of 14, with a score below 14 frequently theorized as being associated with increased risk of injury [32]. Minimal research examining the association between FMS and functional movement exists, although some recent publications have noted a link between stability and functional movement [31,33].

Perceptions of ability across both fundamental and functional movement are considered likely factors in guiding actual movement abilities, either directly or in a mediating capacity [5]. Perceived motor competence (PMC) refers to a person’s own belief in how competently they can execute a particular skill [34]. PMC is formulated through numerous factors, including but not limited to the following: (1) previous experience, (2) task difficulty, (3) persuasion and interaction with others, (4) intrinsic motivation, and (5) vicarious experience [35,36]. PMC is rooted within Harter’s Competence Motivation theory, which suggests that competence (i.e., mastering FMS or functional movement) is related to PMC, with PMC further impacting on PA [36,37]. Further research in motor development has proposed a theoretical model, highlighting a positive spiral of engagement between PMC, FMS (or general motor skills competence), and PA [38].

A systematic review found that PMC and PA have a bidirectional relationship, as they both act as an outcome and a determinant of one another [39], which is further supported across international studies [40,41,42]. Interestingly, PMC has also been found to have an association with actual motor competence (AMC) [43,44]; however, other research has noted a lack of an association between PMC and AMC [40,45]. Regarding PMC and functional movement, O’Brien et al. (2018) observed that Irish adolescents (*N* = 219) have rather high perceptions of ability, in spite of their low actual levels of functional movement [22]. Notably, these high perceptions of ability are in contrast to the theoretical expectations of lower and more realistic perception scores emerging in adolescence due to cognitive maturity [38,46].

There is a glaring need to investigate the extent of the relationship between PMC, FMS, and functional movement. A greater understanding of this relationship could spur the development of effective intervention programs centered on PA, FMS, or functional movement for adolescent populations. The purpose of this study was to gather data on Irish adolescents’ fundamental and functional movement, as well as PMC, to determine if there was an age-related association between variables (i.e., were perceptions of ability becoming more accurate in older participants). PMC was measured, as the developmental model of motor competence indicates that PMC and AMC have a reciprocal relationship, with positive PMC resulting in improvements in MC and vice versa [38]. Furthermore, this model theorizes that perceptions of movement will become more realistic as children progress through adolescence [38].

## 2. Materials and Methods

### 2.1. Study Design and Participants or Participants and Setting

These data were collected at baseline as part of a larger randomized controlled trial study of Irish adolescent youth focused on AMC and PMC. Ethical approval was granted by the Social Research Ethics Committee at University College Cork in November 2018 (UCC Ethics log code “Log 2018-169”). Prior to commencing this school-based study, the leading researchers visited the school principal and liaison PE teacher in each participating school, where a full briefing of the data collection process was outlined. All schools partaking in the study had a qualified post-primary PE teacher, as recognized by the Teaching Council of Ireland, teaching a consecutive double period (80 min of class time) of PE to the participating class groups. While the quality of the existing PE program was not assessed, the provision of qualified teachers and equal time for PE in each school indicates that the teaching experiences for the participating students were relatively equal at baseline. All students who participated came through the Irish primary school system where a generalist classroom teacher led their PE classes. Subsequent to the granted approval from school principals, consent forms and information sheets were distributed to each respective class group. Informed parental consent and child assent were required to be eligible for participation in the study. Both schools and participants were informed that their participation was entirely voluntary, and they were free to withdraw from participation of their own volition at any time. In terms of the research rigor inherent to school-based measurements, it is important to note that the principal investigators for this study were qualified post-primary (second-level) specialist (PE) teachers, as recognized by the Teaching Council of Ireland.

Consenting second-level participants from years one to three (age range: 12.23–16.37 years), across 6 schools, within suburban areas of County Cork, in the province of Munster, Ireland were invited to participate. Two all-male second-level schools, two all-female second-level schools, and two mixed lower socioeconomic status (SES) schools took part, with each pairing matched by gender and SES. This matching protocol should ensure that the equality of PA experiences outside of schooling was relatively similar for students in the different SES bands, although qualitative or self-report data on accumulated PA or sporting experience outside of schooling was not collected. A total of 486 students were eligible to partake in the study, with final consent and attendance provided by 373 participants (14.38 ± 0.87 years; 76.7% of full sample). Of the participants, 178 were female (47.7%) and 195 were male (52.3%), with 101 participants from 1st year (i.e., their first year of second-level schooling), 149 from 2nd year, and 123 from 3rd year.

### 2.2. Collection Methods

Prior to data collection, all field staff, who were undergraduate pre-service or graduate PE teachers, underwent a thorough 3-hour field researcher training workshop on the measurement protocol associated with FMS, FMS^TM^, self-report questionnaires, and body composition. This involved an objective criteria informed process to ensure that field staff applied consistency to their administration and implementation of the respective FMS and functional movement pattern(s) they were assigned. Research assistants practiced their assigned skills at the training workshop, and prior to the arrival of study participants on the day of testing, with their procedures evaluated by the principal investigators during these periods. Additionally, field staff were given a descriptive research handbook detailing the protocol for them to revise prior to all data collection phases. Baseline data were collected with participants in their respective class groups (maximum *n* = 27) during allocated school timetable visits. All measurements (FMS, FMS^TM^, and self-report questionnaires) were carried out during a typical PE class (duration range 80 to 120 min).

### 2.3. Measures

For both the FMS and FMS^TM^ objective measurements, Apple iPads placed on a fixed stanchion were used to record each participant’s performance, and execution of the requisite skill/movement patterns. Camera positioning and angles were kept consistent at all times to ensure that total body movement was captured [11,17]. While live-scoring has proven to be an acceptable method for both inter-rater and intra-rater reliability of the FMS^TM^ [47], video recording was used to allow for more careful data processing at a later time in the university lab [17,48]. Video recordings are an essential component to motor skills data collection, as they allow for greater scrutiny and accuracy of measurement [49].

#### 2.3.1. Fundamental Movement Skills

The following 10 FMS were assessed: horizontal jump, vertical jump, run, skip (locomotor, maximum score of 34), catch, stationary dribble, overhead throw, two-handed strike, kick (object-control, maximum score of 40), and balance (stability, maximum score of 10) for a composite maximum raw score of 84. Each of the 10 FMS were assessed in conjunction with the observable behavioral components from three testing batteries (Test of Gross Motor Development, Test of Gross Motor Development-2, Victorian Fundamental Motor Skills manual), with established reliability and construct validity [50,51,52]. Prior to participant performance, a trained field staff member provided an accurate demonstration and instruction of the skill to be performed. Procedures delineated in the Test of Gross Motor Development-2 (TGMD-2) examiner’s manual [51] were stringently adhered to within the assessment of the 10 FMS during the PE period. To maintain participant consistency within skill performance, no visual or verbal feedback from any of the trained field stuff was given during testing. Skills were performed on three occasions by participants, including a single familiarization practice and two performance trials, as described in other Irish adolescent movement research [11,17]. The FMS scoring process were completed at a later date by the principal investigators.

#### 2.3.2. Functional Movement Screen

The Functional Movement Screen^TM^ (FMS^TM^) is an evaluation tool that comprises a series of movements designed to assess multiple domains of function and the quality of movement patterns [29,53]. All seven components of the FMS^TM^ were assessed: active straight-leg raise, trunk stability push-up, shoulder mobility, deep squat, rotary stability, hurdle step, and in-line lunge. The test administration measures, instructions, and scoring associated with the standardized version of the test [29,53] were applied to maintain accuracy during testing and scoring [30,54]. Normative FMS^TM^ values have been established in adolescent school-aged children [30]. Trained field staff utilized the pre-determined verbal instructions during testing. During data collection, each participant was video recorded and given three attempts to perform the movement. The principal investigators scored the optimum trial rigorously at a later date, as recommended in the original training workshop. The FMS^TM^ has a scoring range from zero to three, with three being the optimum score [55]. If, at any juncture during the testing, the participant displayed signs of pain or discomfort anywhere on their body, they received a score of zero, and the area of discomfort was noted. An inability to complete the movement is characterized by a score of one. A score of two was awarded when the participant used a compensation in order to perform the movement. A maximum score of three was allocated to the participant where the movement was performed correctly without compensation. Bilateral scores for five (active straight-leg raise, shoulder mobility, rotary stability, in-line lunge, and hurdle step) of the seven functional movements were also recorded to compare the imbalance and dysfunction between the left and right sides of the body. The lowest score for either side of the body within the movement contributed to the final score. For each of the seven screening items, the highest score from the three trials was recorded and used to generate an overall composite FMS^TM^ score, with a maximum value of 21, as part of the established protocol [29,53,56]. FMS^TM^ scoring was completed by the principal investigators at a later juncture.

#### 2.3.3. Perceived Motor Competence

The McGrane et al. (2016) physical self-confidence scale is the first valid and reliable instrument that has been developed to assess physical self-confidence in adolescents, and it is at a skill-specific, FMS proficiency level [57]. This Likert-scale measurement tool has excellent test–retest reliability (r = 0.92) [57]. The scale demonstrates good content and concurrent validity (r = 0.72) [57] when compared to the physical self-perception profile [58]. The physical self-confidence scale [57] was used as an indicator to measure the PMC of participants’ in their FMS proficiency, with the question stem altered very slightly from “how confident are you at performing” to “how well can you perform”, following advice from experts and recent developments in the field [59]. Referring to competence will ensure better alignment with the field of PMC and reduce a problem within PMC research of ‘definitional blurring’, where a lack of clarity and alignment between measurement tools and constructs leads to incorrect assumptions being made [59,60]. Participants completed the PMC questionnaire separate from other participants after they performed the actual motor competence test of a specific skill (i.e., the participant evaluated their throw PMC after completing the TGMD-2 test for the throw skill), returning to watch the demonstration of the next skill. Within this PMC scale, participants were asked to rate their competence at performing 10 FMS, based on a Likert-scale format of 1–10, in which a score of ‘1’ indicated being not competent at all and a score of 10 indicated being very competent. By using the Likert scale and modifying the question stem, this scale ensures a more holistic alignment and accurate measurement of PMC. The 10 FMS selected within this instrument are pivotal to the Irish adolescent sporting culture [3,17]. In the current study, Cronbach’s alpha coefficient showed good internal consistency for the perceived motor competence scale, with a Cronbach alpha coefficient of 0.84 overall, 0.73 for the object control subscale, and 0.80 for the locomotor subscale. Both self-report questionnaires (Perceived FMS competence and perceived functional competence) were collected with participants using pen and paper, and later inputted into a statistical programming software by the research team.

#### 2.3.4. Perceived Functional Movement Competence

A previously developed tool to assess perceived functional movement competence amongst an Irish adolescent population was employed in this study to measure the PMC of participants in their functional movement proficiency [22]. As per the FMS scale outlined above, small modifications were made to refer to ‘how well can you perform’, rather than ‘how confident you are’, which is in accordance with the recommended alignment between PMC and FMS assessment [59]. Participants completed the perceived functional movement competence questionnaire separate from other participants after they performed the actual motor competence test of a specific skill (i.e., the participant evaluated their deep squat perceived competence after completing the FMS^TM^ test for the deep squat), returning to watch the demonstration of the next skill. Similar to the McGrane et al. (2016) protocol, participants were asked to rate their competence at performing the identified seven FMS™ tasks, based on a Likert-scale format of 1–10, in which a score of 1 indicated being not at all competent, and a score of 10 indicated being very competent. As the seven FMS™ tasks are non-sport specific and participants may not be familiar with the movements, a visual image alongside the question was provided, similar to empirically validated pictorial instruments for assessing FMS perceived competence [22,61,62]. Previous test–retest reliability coefficients for this perceived functional movement competence scale ranged from 0.82 to 0.93, which showed that the scores across this instrument were stable over time [22]. Furthermore, the perceived functional movement competence scale in this current study presented with good internal consistency, with an overall Cronbach alpha coefficient of 0.85 reported across the seven items.

### 2.4. Data Processing and Analysis

Prior to the data scoring of fundamental and functional movement, inter- and intra-rater reliability was established on 10% of the dataset. Two raters double-coded 10% of the data to determine intra-rater reliability, and both coded the same 10% of data to determine inter-rater reliability [63]. The two principal investigators were required to reach a minimum of 95% inter-rater agreement for all 10 FMS and seven functional movements. Questionnaire data for PMC were collated and inputted into SPSS version 25.0 (SPSS Inc., Chicago, IL, USA) for Windows.

The FMS, FMS™, and PMC datasets were analyzed using SPSS version 25.0 for Windows. Descriptive statistics and frequencies for FMS and FMS™ at the skill and composite score levels were calculated. For both FMS and FMS^TM^, mastery was defined as the correct performance of all components across two trials [17]. Differences in performance by year group in overall and individual FMS/FMS^TM^, and PMC performances were analyzed using one-way between groups analysis of variance (ANOVA) with post-hoc tests. Pearson product moment correlation coefficients were calculated to analyze the strength and direction of the association between PMC scores and AMC scores across both FMS and FMS^TM^ differentiated by school year group. The strength of the relationship (i.e., small, moderate, large) was determined using the Cohen (1988) guidelines [64]. A correlation coefficient of r = 0.10–0.29 denoted a small correlation, r = 0.30–0.49 denoted a moderate correlation, and r ≥ 0.50 denoting a strong correlation, respectively [64].

## 3. Results

This section may be divided by subheadings. It should provide a concise and precise description of the experimental results, their interpretation, as well as the experimental conclusions that can be drawn.

### 3.1. Fundamental Movement Skills

Across the 10 FMS assessed, no participant successfully mastered all skills. The percentage mastery for each FMS by year group is shown in Figure 1, while the actual FMS proficiency by year group is given in Table 1. The highest performed skill was the run (60.4% mastery in 1st year, 70.3% in 2nd year, and 62.6% mastery in 3rd year, respectively). For 1st and 2nd-year students, the poorest performed skill was the horizontal jump (14% mastery and 7.7% mastery respectively); however, for the 3rd-year cohort, the poorest performed skill was the kick (10.3%).

A one-way ANOVA (Table 1) revealed no significant differences in gross motor FMS competence across the three year groups. At the individual skill level, a significant decrease in horizontal jump proficiency was evident from 1st year (5.19 ± 1.75) to 2nd year only (4.99 ± 1.8) (*p* < 0.001). Participants’ catching proficiency increased significantly from 2nd year (4.5 ± 0.95) to 3rd year (4.94 ± 1.01) (*p* < 0.001).

### 3.2. Functional Movement

No student attained mastery across all seven functional movement assessments (maximum score of 3 per test). Figure 2 displays the prevalence of functional movement mastery among participants by year group. Shoulder mobility was the best performed movement pattern across all three year groups, with 23.8%, 26.9%, and 24% mastery across the 1st, 2nd and 3rd-year groups, respectively. The poorest performed movement pattern was the rotary stability, with 0% of participants obtaining mastery across all three year groups.

When broken down by year group, a one-way ANOVA (Table 1) revealed no significant differences in functional movement performances at the gross or at the individual movement pattern level.

### 3.3. Perceived Motor Competence

A one-way ANOVA broken down by year group revealed no significant differences in PMC levels across the 10 FMS (see Table 2) between year groups.

A one-way ANOVA broken down by year group (Table 2) revealed that participants had significant differences in their perceptions of rotary stability competence across years. Post-hoc tests revealed that there was a significant difference in their perception of this functional movement between 1st year (m = 5.70 ± 2.32) and 2nd-year participants (m = 4.71 ± 2.04).

### 3.4. Relationship Actual Motor Competence (FMS and Functional Movements) and Perceived Motor Competence

#### 3.4.1. FMS

1st Years—A small positive relationship was found between PMC scores for 1st-year participants in overall gross motor FMS competence (r = 0.29, *p* = 0.007), the vertical jump (r = 0.24, *p* = 0.02), throw (r = 0.27, *p* = 0.007), strike (r = 0.29, *p* = 0.005), and the balance (r = 0.27, *p* = 0.006). A moderate positive relationship was found between PMC scores for 1st-year participants in the object control composite subset (r = 0.38, *p* ≤ 0.001) and the kick (r = 0.38, *p* ≤ 0.001).

2nd Years—A small positive correlation was found between PMC scores for 2nd-year participants in overall gross motor competence (r = 0.28, *p* = 0.006), the throw (r = 0.25, *p* = 0.004), strike (r = 0.2, *p* = 0.03), and kick (r = 0.18, *p* = 0.05). A moderate positive correlation was found between PMC scores for 2nd-year participants in the object-control composite subset (r = 0.36, *p* ≤ 0.001) and the balance (r = 0.30, *p* ≤ 0.001).

3rd Years—Small positive correlations were found between PMC scores for 3rd-year participants in the skip (r = 0.24, *p* = 0.008), horizontal jump (r = 0.22, *p* = 0.02), strike (r = 0.27, *p* = 0.003), and the balance (r = 0.29, *p* = 0.001). A moderate positive correlation was found between PMC scores for 3rd-year participants in overall gross motor competence (r = 0.41, p ≤ 0.001), the locomotor composite subset (r = 0.33, *p* ≤ 0.001), the object-control composite subset (r = 0.4, *p* ≤ 0.001), throw (r = 0.35, *p* ≤ 0.001), and kick (r = 0.31, *p* = 0.001).

#### 3.4.2. Functional Movement

1st Years—Small positive correlations were found between PMC scores for 1st-year participants in the overall composite functional movement score (r = 0.25, *p* = 0.03), the in-line lunge (r = 0.28, *p* = 0.008), and the shoulder mobility (r = 0.26, *p* = 0.01). Moderate positive correlations were found between PMC scores for 1st-year participants in the deep squat (r = 0.31, *p* = 0.002) and the trunk stability push up (r = 0.38, *p* ≤ 0.001)

1st Years—Small positive correlations were found between PMC scores for 1st-year participants in the overall composite functional movement score (r = 0.25, *p* = 0.03), the in-line lunge (r = 0.28, *p* = 0.008), and the shoulder mobility (r = 0.26, *p* = 0.01). Moderate positive correlations were found between PMC scores for 1st-year participants in the deep squat (r = 0.31, *p* = 0.002) and the trunk stability push up (r = 0.38, *p* ≤ 0.001)

2nd Years—Small positive correlations were found between PMC scores for 2nd-year participants in the active straight-leg raise (r = 0.23, *p* = 0.008), deep squat (r = 0.2, *p* = 0.03), hurdle step (r = 0.23, *p* = 0.01), shoulder mobility (r = 0.23, *p* = 0.01), the trunk stability push-up (r = 0.23, *p* = 0.01), and the overall composite functional movement score (r = 0.26, *p* = 0.01).

3rd Years—Small positive correlations were found between PMC scores for 3rd-year participants in the hurdle step (r = 0.2, *p* = 0.03) and the trunk stability push-up (r = 0.28, *p* = 0.003). Moderate positive correlations were found between PMC scores for 3rd-year participants in the shoulder mobility assessment (r = 0.37, *p* ≤ 0.001).

## 4. Discussion

The intention of this study was to derive findings on FMS, FMS^TM^, and PMC to see if an age-related association existed among adolescents. To the author’s knowledge, this is only the second study of its kind to combine both actual and perceived fundamental and functional movement assessments among a general adolescent population. Within the current study, no participant across the age range of 12–16 years old demonstrated overall mastery across all 17 assessed FMS and functional movements. This overall lack of proficiency may suggest that Irish children and adolescents are transitioning to more sport-specific games and skills, without acquiring basic motor competencies [17,65]. These movement skill and pattern deficiencies may also help explain the steep decline in both PA and community sport participation among Irish teenagers [3,66], as adolescents may be ‘dropping out’ due to lower skill levels. The high and misaligned perceptions of movement do not appear to be a mediating factor on skill ability; high perceptions that are not reflective of actual ability may in fact be a contributing factor to activity withdrawal, as adolescents look to maintain the illusion of ability to themselves and their peers [67].

Data from Figure 1 would suggest low overall FMS proficiency among participants. The mastery of object-control skills in this study was demonstrably lower than previously published figures among Irish adolescents [17,20] and Irish children [18,19]. While performance in the locomotor skills of the run and horizontal jump were below previous Irish standards, both the vertical jump and skip were comparable or higher than other Irish childhood and adolescent research [17,18,19,20,68]. In comparison with global studies, performances in the kick, dribble, and balance were lower than previously reported values in British children [69]. Participant performance in both the horizontal jump and catch from this study were lower than the previously reported values among Belgian 8-year-old children [70]. In terms of overall gross motor skill proficiency, all FMS scores in the current study were lower than published values among Australian children [71].

The values from Table 1 indicate that there is no significant age-related association in the growth of FMS. Older individuals often exhibit improved FMS due to increased access to practice, coaching, corrective feedback, and greater physical stature; however, FMS improvements are not always age-determined [6,72]. This study’s findings are supported by previous Irish research that found an age-related decline in object-control proficiency across the second-level school years [20]. More recent Irish findings indicate a ‘plateau’ of skills from the age of 10, with no further growth in FMS for children aged 11 and 12 [19]. This is aligned with critical principles of motor development, which suggest that motor skills require purposeful practice and are not a product of maturity [13,73].

This troublesome lack of growth in FMS will be detrimental to long-term PA and global health, given the aforementioned association between FMS and PA in addition to other markers of health [38,74]. Seefeldt’s motor proficiency barrier theory similarly affirms this by noting how those who fail to become competent in motor skills will disengage from PA and demonstrate lower physical fitness capacity [75]. This theory has been affirmed in young adult populations [76], and recent research on a cohort of Belgian children has suggested that this barrier exists in later childhood [77].

This study presented lower overall composite and individual functional movement values when compared to previous research conducted among Irish adolescents [20,22] and global adolescent populations [23,30,31]. The overall mean value of approximately 12 across the three year groups is lower than the score of 14. This score of ‘14’ or lower has previously been cited as evidence of movement dysfunction, which may increase propensity for injury [32,78]. The lack of any significant age-related effects on functional movement is notable (Table 1), given the fact that age has been cited as being a moderate predictor of performance on the FMS^TM^ among childhood populations [79]. The lack of strong functional movement capacity may be due to additional factors, such as postural instability, resulting from extended static periods during schooling [20]. As functional movement has been linked with improvements in PA behavior [56,80], these low values are troublesome and indicate it is imperative to focus on growing functional capacities in PE and sporting environments to promote efficient and safe movement among adolescent populations. Functional movement exemplifies the definition of MC, as presented by Robinson et al. (2015), in addition to its theorized links with FMS skills [26,81]. The strength and mobility elements of the FMS^TM^ typify movements that are often unaccounted for in MC testing batteries but warrant inclusion in a broader definition of MC [82].

Table 2 highlights the high PMC values for FMS across all three year groups. All 10 FMS presented with a mean PMC value above 6 (out of 10). While lower than previously documented Irish values of PMC [22,57], Table 2 further highlights the high PMC values and the lack of significant change in these PMC values across different year groups. A comparable study in Belgium presented moderate relationships between PMC and AMC [83]. While in the current study, the correlational analysis between object control PMC and object control AMC met the moderate threshold across all years, no other composite score (i.e., overall gross motor or locomotor) or skill did, suggesting that Irish adolescents may possess lower accuracy of assessment compared to their Belgian counterparts. The 3rd-year group did possess slightly more moderate correlations than both other year groups in terms of AMC and PMC associations; however, as 2nd years did not possess more moderate or small significant correlations than 1st years these findings suggest that there is little consistency in cognitive maturity, and that more realistic and accurate perceptions do not emerge as adolescents age, as has been previously theorized [38,46].

Regarding actual functional movements and PMC alignment (Table 2), lower, aligned, and realistic PMC functional movement values were expected, as the success criteria for performance were shared with participants during assessment, in accordance with the FMS^TM^ protocol [29]. As such, participants may have a greater understanding of what constitutes successful performance for the specific functional movement patterns, and therefore, they may be more aware when evaluating their work [84,85]. Mean perceived functional movement scores over 6 (out of 10) were found for all patterns, bar the rotary stability. These perceived functional movement values were lower than previous values in the literature [22]; however, there were no significant changes across year groups between actual functional movement proficiency and PMC scores, albeit with the exception of rotary stability. A significant change was found in rotary stability between year groups, with second years possessing significantly lower actual and perceived values than 1st years and 3rd years, respectively. An expected decline in PMC values with aging is expected [38], which did not generally occur to a statistically significant level across the other movements.

The theoretical assumption of cognitive growth [38,46] suggests that a greater degree of alignment between PMC and AMC should emerge across the adolescent years with greater correlation coefficients emerging across each year group. The ‘veridicality’ of PMC, i.e., the degree of alignment between perception of ability and actual ability [86] does not appear to show linear improvement as participants age, contrary to conventional theories in the field of PMC [43,46]. The lack of improvement in the veridicality in this study appears to be consistent across PMC and FMS assessment, and PMC and functional movement assessment, respectively.

This study’s findings as well as those of previous Irish [22,57,87] and international work [34,43,88] suggest that high PMC values are common in later childhood and early adolescence. Furthermore, these values may even remain stable with aging; a study of Dutch and German children (*n* = 198; male = 110; female = 88; mean age = 9.35 ± 0.67 years) found no significant differences in their perception of ability a year after their initial assessment, with values still remaining high [34]. This potentially indicates that the theoretical assumption of greater accuracy of assessing one’s PMC as children age may be flawed. The reasons for this are innumerable; it has been suggested that enhanced awareness and social comparisons of their peers may interfere with self-assessment among adolescents [87]. The potential for social desirability bias impacting on self-report data should be noted; this is a common problem in PA research, where a disconnect between reported behavior and actual behavior exists [89,90].

A veridical, i.e., aligned and accurate perception, is more desirable as recent research indicates that accuracy in the assessment of PMC may be associated with engagement in PA [88]. Harter’s work within competence motivation theory suggests that high perceptions are quite important, but actual ability and realistic perceptions must remain a key focus when working with youths [91]. Interventions designed to boost PMC and nurture accuracy of perception often prioritize developmentally appropriate exercises that focus on FMS deficiencies and enable all participants to achieve success at a level commensurate with their abilities [92,93]. A similar approach could be advocated for FMS^TM^ exercises, given that improvements in functional capacity have been shown in as little as six weeks when incorporated into warm-up exercises for students [23]. Furthermore, increased knowledge of the FMS^TM^ testing criteria, as would happen in an intervention with a specific purpose and focus on improving functional movement, has been shown to boost FMS^TM^ capacity [94]. Improvements in MC, veridicality, and levels of perception have been achieved through effective pedagogy styles taking a student-centered approach (i.e., purposeful, student-led goals, opportunities for reflection, positive reinforcement, appropriately challenging tasks) [67,86].

A focus on skill improvements through student-centered activities that promote engagement and understanding are pivotal to increasing the AMC and accuracy of PMC among childhood and adolescent populations [67,86,92]. The student-centered approach will nurture actual skill improvements, while promoting the cognitive engagement necessary for students to form a better understanding of skill performances, potentially resulting in more realistic and accurate PMC scores [86,95]. Veridical and realistic perceptions of ability are critical to accruing PA and improving movement quality, as they are associated with the sustained motivation and fulfillment of goals; programs designed to ameliorate deficiencies in AMC and PMC must take these ideas into account [86,88,96]. Reducing actual skill deficiencies in addition to promoting veridical perceptions must become a central tenet of future movement programs predicated on motor development principles.

### Strengths and Limitations

A potential limitation of this study is that it is cross-sectional in nature. While the sample size is large and data were collected from a representative mixture of single sex, mixed-sex, and disadvantaged area school populations, all participants were selected from one county in the south of Ireland, which may not be representative of Irish adolescents in rural settings or other regions. The use of valid and reliable tools for both perceived and actual motor competence is a strength of the study; however, self-report data of perceived motor and perceived functional competence may result in positive response bias or overestimation. The current study does not provide evidence of a causal relationship between motor competence and perceived motor competence. To gain greater insight into this relationship and how it evolves as individuals age, longitudinal or intervention studies with a greater sample size are required.

## 5. Conclusions

The low level of AMC in both fundamental and functional movement among the Irish adolescent population indicates the need for developmentally appropriate activities and effective teaching strategies that would ameliorate these movement deficiencies. Continued focus on FMS in both primary school and across the early second-level school years may be necessary, as deficiencies in FMS are apparent throughout the first three years of second-level school. The use of specialist PE teachers in primary school PE or in after-school PE programs at the primary level may need to be considered to ensure students enter the second level with appropriate FMS ability, since the current practice of generalist classroom teachers teaching PE has proved ineffective in developing FMS. High levels of PMC across both FMS and functional movement exist; however, there does not appear to be a strong association between PMC and FMS or functional movement among Irish adolescents, nor does there appear to be an age-related association between PMC and AMC. Teaching strategies that clearly elucidate to students how to execute movement skills and patterns will be critical to improving their actual abilities in addition to their cognitive capacities and perception of their abilities. Future interventions will need to focus on addressing the movement deficit that is abundantly evident in adolescents and further creating more realistic perceptions of ability among Irish adolescent youths. Further studies examining the association between PMC and AMC are warranted to explore the presence or absence of cognitive growth as children age, as this current study found no substantial age-related link between PMC and AMC.

## Figures and Tables

**Figure 1 ijerph-17-06361-f001:**
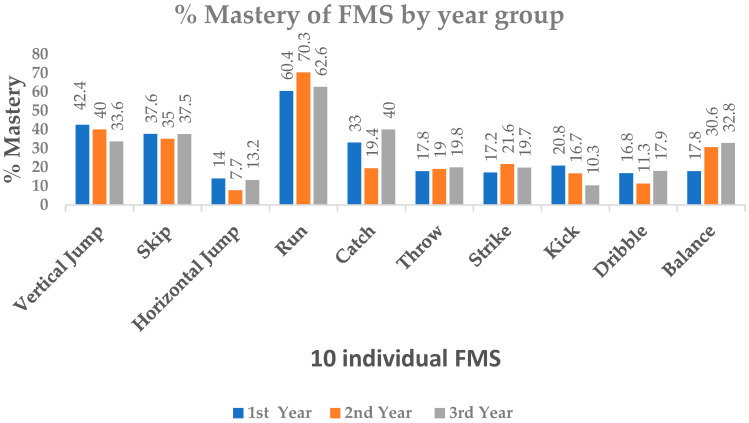
The prevalence of fundamental movement skills (FMS) mastery by school year group among participants.

**Figure 2 ijerph-17-06361-f002:**
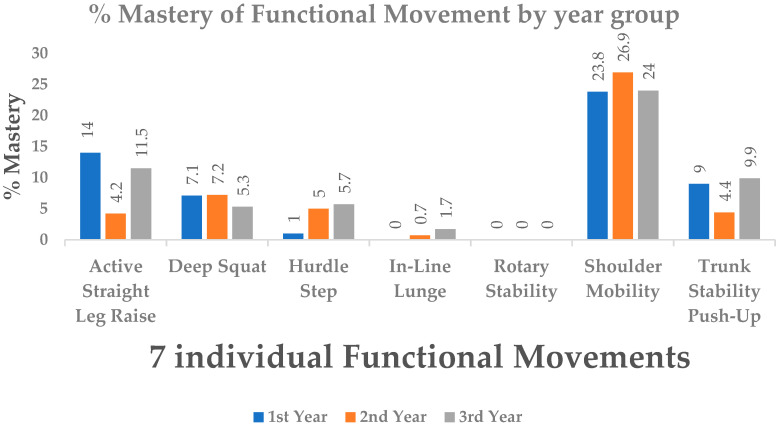
The prevalence of functional movement mastery by school year group among participants.

**Table 1 ijerph-17-06361-t001:** Actual FMS and FMS^TM^ proficiency by year group (ANOVA results).

**FMS Skills (Max Raw Score)**	**Year 1** **(Mean ± SD)**	**Year 2** **(Mean ± SD)**	**Year 3** **(Mean ± SD)**	***p* Value**	**Post-Hoc Tests**
Gross Motor Skill (84)	65.03 ± 6.43	64.73 ± 5.99	64.98 ± 6.28	0.93	N/A
Locomotor Skill (34)	27.55 ± 3.66	26.85 ± 3.43	26.79 ± 3.71	0.24	N/A
Object Control Skill (40)	29.53 ± 4.24	29.58± 3.67	29.90 ± 3.85	0.75	N/A
Vertical Jump (12)	10.40 ± 1.82	10.2 ± 1.91	10.14 ± 1.87	0.56	N/A
Skip (6)	4.69 ± 1.24	4.43 ± 1.44	4.52 ± 1.45	0.34	N/A
Horizontal Jump (8)	5.19 ± 1.75	4.64 ± 1.58	4.99 ± 1.8	0.04 *	Y1 < Y2
Run (8)	7.28 ± 0.99	7.38 ± 1.11	7.13 ± 1.27	0.20	N/A
Catch (6)	4.8 ± 0.98	4.5 ± 0.95	4.94 ± 1.01	0.001 *	Y2 < Y3
Throw (8)	5.18 ± 2.1	5.35 ± 2.06	5.1 ± 2.19	0.60	N/A
Strike (10)	7.66 ± 1.77	8.08 ± 1.53	7.77 ± 1.64	0.10	N/A
Kick (8)	6.15 ± 1.31	5.85 ± 1.38	5.74 ± 1.32	0.08	N/A
Dribble (8)	5.75 ± 1.53	5.80 ± 1.23	6.05 ± 1.29	0.20	N/A
Balance (10)	7.92 ± 1.66	8.23 ± 1.73	8.22 ± 1.65	0.30	N/A
**Functional movements**	**Year 1** **(Mean ± SD)**	**Year 2** **(Mean ± SD)**	**Year 3** **(Mean ± SD)**	***p* Value**	**Post-Hoc**
Active Straight Leg Raise (3)	1.72 ± 0.70	1.66 ± 0.56	1.69 ± 0.67	0.78	N/A
Deep Squat (3)	1.48 ± 0.63	1.52 ± 0.63	1.48 ± 0.60	0.82	N/A
Hurdle Step (3)	1.69 ± 0.49	1.73 ± 0.55	1.80 ± 0.53	0.34	N/A
In-Line Lunge (3)	1.69 ± 0.46	1.74 ± 0.45	1.78 ± 0.46	0.41	N/A
Rotary Stability (3)	1.72 ± 0.45	1.84 ± 0.37	1.80 ± 0.40	0.09	N/A
Shoulder Mobility (3)	1.94 ± 0.73	2.06 ± 0.69	1.95 ± 0.73	0.31	N/A
Trunk Stability Push Up (3)	1.47 ± 0.66	1.45 ± 0.58	1.63 ± 0.66	0.06	N/A
FMSTM Overall (21)	11.78 ± 1.19	12.06 ± 1.88	12.14 ± 2.12	0.40	N/A

* The mean difference is significant at the 0.05 level. N/A = Not Applicable. FMS^TM^ = Functional Movement Screen^TM^.

**Table 2 ijerph-17-06361-t002:** Perceived motor competence (PMC) of the 10 FMS and 7 functional movements by year group.

**FMS PMC** **Rating**	**1st Year** **(Mean ± SD)**	**2nd Year** **(Mean ± SD)**	**3rd Year** **(Mean ± SD)**	**Year Group** **(*p* Value)**	**Post-Hoc Tests**
Vertical Jump (10)	7.27 ± 2.11	6.97 ± 1.94	7.11 ± 1.78	0.52	N/A
Skip (10)	8.05 ± 2.14	7.70 ± 1.9	7.74 ± 2.02	0.38	N/A
Horizontal Jump (10)	6.84 ± 2.13	6.36 ± 2.06	6.34 ± 2.09	0.15	N/A
Run (10)	8.44 ± 1.67	8.09 ± 1.65	8.19 ± 1.64	0.28	N/A
Catch (10)	8.98 ± 1.44	8.95 ± 1.39	8.72 ± 1.76	0.38	N/A
Throw (10)	7.65 ± 1.91	7.37 ± 1.85	7.82 ± 1.70	0.15	N/A
Strike (10)	6.23 ± 2.85	6.20 ± 2.46	6.42 ± 2.41	0.77	N/A
Kick (10)	7.40 ± 2.24	7.18 ± 2.23	7.48 ± 1.86	0.50	N/A
Dribble (10)	8.09 ± 1.89	7.87 ± 1.85	8.10 ± 1.5	0.53	N/A
Balance (10)	8.46 ± 1.76	8.31 ± 1.55	8.40 ± 1.73	0.81	N/A
**Functional Movement PMC Rating**	**Year 1** **(Mean ± SD)**	**Year 2** **(Mean ± SD)**	**Year 3** **(Mean ± SD)**	**Year Group** **(*p* value)**	**Post Hoc**
Active Straight Leg Raise (10)	6.90 ± 2.11	6.63 ± 6.76	6.76 ± 2.10	0.63	N/A
Deep Squat (10)	6.95 ± 2.24	6.91 ± 2.15	6.84 ± 2.13	0.94	N/A
Hurdle Step (10)	6.86 ± 1.96	6.71 ± 2.14	6.88 ± 2.14	0.76	N/A
In-Line Lunge (10)	7.27 ± 1.87	7.11 ± 1.840	6.90 ± 2.11	0.38	N/A
Rotary Stability (10)	5.69 ± 2.32	4.71 ± 2.04	5.14 ± 2.05	0.003 *	Y1 > Y2 < Y3
Shoulder Mobility (10)	6.86 ± 2.33	6.55 ± 2.20	6.32 ± 2.07	0.20	N/A
Trunk Stability Push-Up (10)	6.93 ± 2.38	6.43 ± 2.39	6.53 ± 2.38	0.29	N/A

* The mean difference is significant at the 0.05 level.

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
