# Peer review of "Investigating the Age-Related Association between Perceived Motor Competence and Actual Motor Competence in Adolescence"

_ijerph, 2020, doi:10.3390/ijerph17176361_

Round 1

Reviewer 1 Report

The entire article focuses on "adolescence", so the authors should review the title, since it speaks of "early adolescent youth".

line 126, is there a clinicaltrials.org or similar registry?

line 152, did you perform a test phase, before evaluating?

line 217, this instrument was applied before or after the other tests?, clarify

line 343, letter "w" does not correspond

line 443, does not close parentheses

line 444, double-space before "appropriately"

discussion, add limitations and strengths of the study

Reviewer 2 Report

the research design, methods, number of subjects, introductions explanation of the study are all sound. my concern and not sure if I missed something but these are my concerns.

  • there were boys from all boys schools and girls from all girls school
    • Question what was the quality of physical education in both the schools?
    • was the quality the same or different
  • What is the background of the students did they come from enriched background where parents had them involved in out of school physical activities?
  • These fundamental motor skills are skills that are taught a lower grades ages 5 years old to 10years old
    • what was the quality of the physical education at the primary grade levels to lead up to skills- where these skills should of been taught?

Yes as pointed out more emphasis seems to be more on sport skills and that is what should be taught at that grade level

If I didn't teach at a middle school in NM where the children in the primary grades are taught physical education by classroom teacher and saw the results of when they got to us at the middle school had poor motor skills because of th quality of the program they came from.

this needs to be explained to me in the introduction as one of the reasons why they may have poor basic motor skills. And to me mentioned in the conclusion for future studies of what the skills were coming into program
